# Turbulent hydrodynamics in strongly correlated Kagome metals

Domenico Di Sante [1], Johanna Erdmenger[1], Martin Greiter [1], Ioannis Matthaiakakis[1], René Meyer[1],
David Rodríguez Fernández[1], Ronny Thomale [1✉], Erik van Loon[2] & Tim Wehling[2]

A current challenge in condensed matter physics is the realization of strongly correlated, viscous electron fluids. These fluids can be described by holography, that is, by mapping them onto a weakly curved gravitational theory via gauge/gravity duality. The canonical system considered for realizations has been graphene. In this work, we show that Kagome systems with electron fillings adjusted to the Dirac nodes provide a much more compelling platform for realizations of viscous electron fluids, including non-linear effects such as turbulence. In particular, we find that in Scandium Herbertsmithite, the fine-structure constant, which measures the effective Coulomb interaction, is enhanced by a factor of about 3.2 as compared to graphene. We employ holography to estimate the ratio of the shear viscosity over the entropy density in Sc-Herbertsmithite, and find it about three times smaller than in graphene. These findings put the turbulent flow regime described by holography within the reach of experiments.

[1] Institut für Theoretische Physik und Astrophysik and Würzburg-Dresden Cluster of Excellence ct.qmat, Julius-Maximilians-Universität Würzburg, Am Hubland, 97074 Würzburg, Germany. [2] Institut für Theoretische Physik, Universität Bremen, Otto-Hahn-Allee 1, 28359 Bremen, Germany.
✉email: rthomale@physik.uni-wuerzburg.de

Electrons in solids typically interact not only with impurities and phonons, but also with each other via the Coulomb interaction. If the momentum relaxing effects of impurities and phonons are weak, the Coulomb interaction can become dominant and lead to local thermalization and the formation of an electronic fluid[1,2]. Thus, the length and time scales over which thermalization occurs are controlled by the strength of the Coulomb interaction. The regime of electron hydrodynamics has been realized in several systems[3–7], giving rise to new transport properties[8–12] distinctly different from the ballistic regime. Therefore, characterizing the transport properties of viscous electronic fluids is tantamount to determining the strength of the Coulomb coupling $\alpha$.

The Coulomb interaction also mediates energy and momentum transfer within the fluid ensuring a local thermal equilibrium, and thereby controls transport coefficients such as the shear viscosity $\eta$. Since direct access to the Coulomb coupling $\alpha$ proves difficult, we focus in this work on the easily accessible shear viscosity, or more precisely on the ratio between $\eta$ and the entropy density of the fluid, $s$. As explained in the "Methods" section, the shear viscosity $\eta$ is straightforwardly obtained from a Kubo formula (see Eq. (16) in "Methods" section), and the entropy density $s$ from thermodynamics (see Eq. (8) in "Methods" section). For a Dirac fluid, the ratio $\eta/s$ is equal to the temperature multiplied by the kinematic viscosity $\nu$, which is experimentally accessible[11,12], $\eta/s = T\nu$. The physical relevance of $\eta/s$ is that it yields the shear viscosity per effective degree of freedom participating in the momentum diffusion in the fluid. The ratio $\eta/s$ depends sensitively on $\alpha$, as shown in Fig. 1. In the weakly Coulomb interacting regime, i.e. for $\alpha \ll 1$, first order perturbation theory such as Boltzmann's kinetic theory predicts a fast fall off, $\eta/s \sim \alpha^{-2}$, as shown by the black line in Fig. 1. For intermediate Coulomb couplings, perturbative approaches lose their validity. Holographic gauge/gravity duality[13] provides a nonperturbative approach to predict the coupling dependence of $\eta/s$. In the limit of infinitely strong coupling and for systems with a large number of degrees of freedom, it predicts the universal value[14–16] (see the red dashed line in Fig. 1)

$$\frac{\eta}{s} = \frac{1}{4\pi} \frac{\hbar}{k_B}. \tag{1}$$

The essential feature of this result is that it is significantly smaller than any value of $\eta/s$ obtained within weak-coupling perturbation theory. Beyond the infinite coupling limit, gauge/gravity duality allows to include finite-coupling corrections to the infinite coupling result in an expansion in the inverse coupling. As we explain in the "Methods" section, the leading-order finite-coupling correction relevant to the materials considered is

$$\frac{\eta}{s} = \frac{\hbar}{4\pi k_B}\left(1 + \frac{\mathcal{C}}{\alpha^{3/2}}\right), \tag{2}$$

where $\alpha$ is the fine-structure constant and the constant $\mathcal{C}$ parametrizes the class of gauge/gravity duals considered.

So far, the material of choice to investigate hydrodynamics of electronic systems has been graphene[17]. Here we show that in certain Kagome materials, hydrodynamic behavior will be significantly enhanced. Specifically, we focus on Kagome materials at filling levels such that the chemical potential is located at the Dirac point. These materials are particularly suited for hydrodynamic studies since not only the Coulomb interaction is enhanced as compared to graphene (see below), but also because the Kagome lattice structure suppresses the formation of ordered phases. The reason is that, in contrast to graphene, the Dirac cones on the Kagome lattice are located far away from half filling. As explained in the "Methods" section, combined with the small low energy density of states, this suggests a strong resilience of the metallic Dirac state against ordering instabilities[18,19], which implies that it sustains stronger Coulomb coupling than a Dirac metal on the honeycomb lattice[20]. For the explicit candidate Kagome metal Scandium-substituted Herbertsmithite $ScCu_3(OH)_6Cl_2$ (Sc-Herbertsmithite hereafter, see Fig. 2)[21], we further calculate the phonon spectrum and find that the optical phonons decouple from the electronic degrees of freedom for temperatures below ~80 K, ensuring that a Kagome Dirac metal with electronic interactions is the appropriate microscopic description.

## Results

**The fine-structure constant.** The suppression of gapped ordered phases and the decoupling of phonons in the proposed Kagome materials allows the electrons to form a fluid by Coulomb interactions up to very low temperatures. The electronic Dirac fluid of particle-hole excitations around the Dirac point then has emergent relativistic symmetry, is parity and time reversal invariant, and particle-hole symmetric. With these symmetries in place, a relativistic fluid is described by relativistic hydrodynamic equations of motion, which depends on the following key parameters: The Fermi velocity $v_F$ of the relativistic dispersion relation playing the role of the speed of light, the relative dielectric constant $\epsilon_r$ in the medium, and the shear viscosity $\eta$ of the fluid. As explained in the "Methods" section, the two other transport coefficients bulk viscosity $\xi$ and interaction induced conductivity $\sigma_Q$ are not relevant for our arguments. The quantities $v_F$ and $\epsilon_r$ set the value of the fine-structure constant (effective Coulomb coupling) $\alpha$ via

$$\alpha = \frac{e^2}{\epsilon_0 \epsilon_r \hbar v_F}, \tag{3}$$

where $\epsilon_0$ is the dielectric constant in vacuum. In turn, $\eta$ depends on $\alpha$. We calculate $v_F$ and $\epsilon_r$, and hence $\alpha$, within the framework of the constrained Random Phase Approximation (cRPA) (see the "Methods" section), and compare the results for

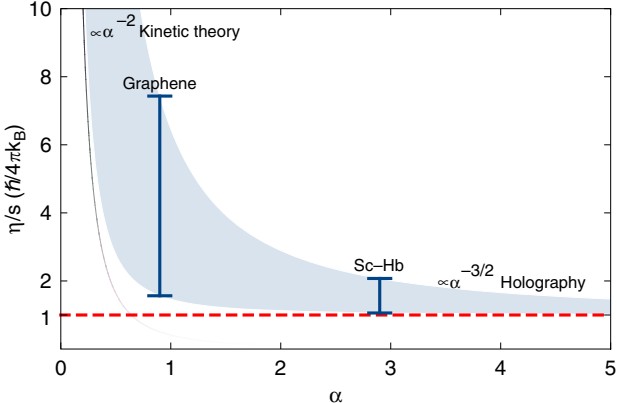

**Fig. 1 $\eta/s$ as a function of the coupling strength.** Black line: prediction in the weak coupling regime, $(\eta/s) \propto \alpha^{-2}$, reliable for small values of $\alpha$. The red dashed line corresponds to the universal holographic value, $\eta/s = \hbar/4\pi k_B$. The blue shaded region, for which $\eta/s$ is given by Eq. (2), represents the holographic prediction. This shaded region parametrizes the class of extrapolating models beyond the $\alpha \to \infty$ limit (see "Methods" section). Notice that Sc-Herbertsmithite (Sc-Hb) shows a much smaller variance than graphene (given by the vertical blue bars at $\alpha = 2.9$ for Sc-Hb and $\alpha = 0.9$ as a representative value for graphene, respectively), providing further support for the applicability of holographic methods to correlated Kagome metals.

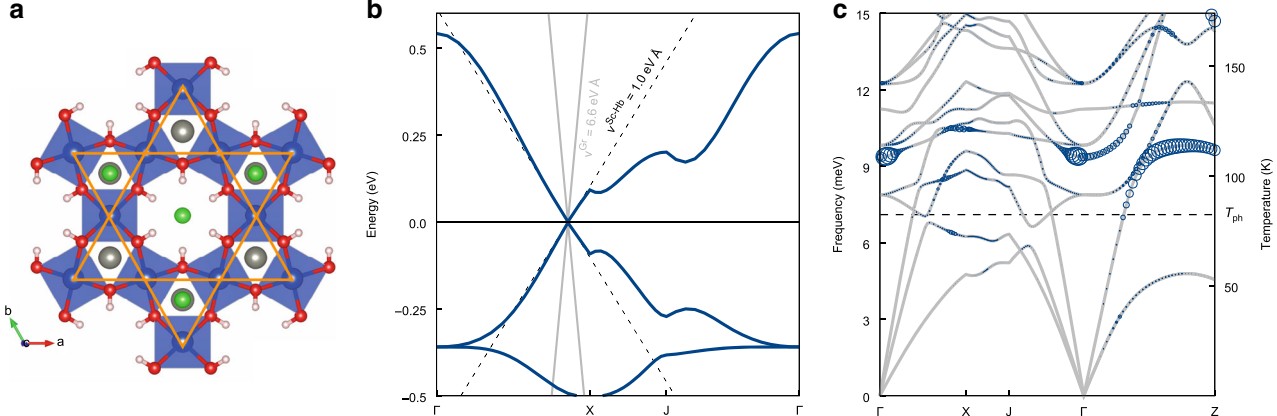

**Fig. 2 First-principles analysis of Sc-Herbertsmithite. a** Top view of the crystal structure, where the $CuO_4$ plaquettes form a Kagome lattice, as highlighted by the orange lines. **b** Ab-initio band structure of the low-energy manifold of Sc-Herbertsmithite along the high-symmetry directions of the conventional hexagonal Brillouin zone[21]. The dashed black lines refer to a linear fit around the Dirac point of Sc-Herbertsmithite, whereas the solid gray lines denote the counterpart for graphene. **c** Phonon dispersion of Sc-Herbertsmithite (gray lines) and relative distribution of the electron-phonon coupling strengths $\lambda_{\nu\mathbf{q}}$ (blue circles, $\nu$ is the branch index, $\mathbf{q}$ the phonon momentum). The horizontal dashed line marks the temperature $T_{ph}$ above which optical phonon modes with sizeable $\lambda_{\nu\mathbf{q}}$ are thermally activated.

**Table 1 Dirac fluid parameters.**

|  | $v_F$ (eVÅ) | $\epsilon_r$ | $\alpha = e^2/\epsilon_0\epsilon_r\hbar v_F$ |
|---|---|---|---|
| ED in vacuum | $2 \times 10^3$ | 1 | 1/137 |
| hBN/graphene/hBN | 6.6 | 2.2–4.0 | 0.5–1.0 |
| Sc-Herbertsmithite | 1.0 | 5.0 ± 0.5 | 2.9 ± 0.3 |

Fermi velocity $v_F$, relative dielectric constant $\epsilon_r$ and fine-structure constant $\alpha$ for electrodynamics in vacuum, (hBN encapsulated) graphene and stochiometric Scandium-substituted Herbertsmithite.

Sc-Herbertsmithite with (hBN encapsulated) graphene. While our findings are likely to be applicable to a broad class of Kagome metals, Sc-Herbertsmithite suitably underlines a prime motif of how to accomplish Dirac fillings in Kagome systems. In pristine Herbertsmithite, $Zn^{2+}$ acts as a charge donor to the Cu $d_{x^2-y^2}$ orbitals which dominate the low energy theory, yielding a half filled Kagome lattice setting. Synthesis of the otherwise identical compound with $Zn^{2+}$ replaced by $Sc^{3+}$ provides the precise stoichiometry for the electrochemical potential to coincide with the Dirac points. While our ab-initio studies can only confirm the chemical stability of the ultimate Sc-Herbertsmithite crystal, the similar atomic radius of Sc in comparison to Zn at least opens the possibility that a related chemical synthesis may be possible. Note that in ref. [21] the similar motif was pursued with Ga-substituted Herbertsmithite. For chemical reaons, we believe Sc to be more suited, because Ga quickly turns fluid and may more easily leak out within a chemical synthesis procedure at higher temperature.

Our first result (c.f. Table 1) is that the fine-structure constant $\alpha$ in Sc-Herbertsmithite is more than three times larger than in graphene, implying a strong enhancement of applicability of viscous hydrodynamics in this material. With Sc-Herbertsmithite being our candidate material for the realization of holographic hydrodynamics, we refer to graphene as a benchmark for our theoretical analysis, as well as a reference point at weaker coupling, which helps us to identify expected trends in observables in more strongly coupled Dirac materials.

**Corrections to the infinite coupling result.** Our second result is an estimate of the leading finite coupling correction in Eq. (2)

within a broad class of holographic models, as explained below and in the "Methods" section. This estimate leads to the blue band in Fig. 1, allowing for predictions for the possible range of $\eta/s$ for both graphene and Sc-Herbertsmithite. Our predictions are indicated by the blue bars and show not only a considerably smaller value $\eta/s$ for Sc-Herbertsmithite as compared to graphene, but also a smaller variance. In particular, our holographic estimate shows that $\eta/s$ for the Dirac fluid in Sc-Herbertsmithite lies significantly closer to the infinite coupling value of Eq. (1) predicted by gauge/gravity duality, than in graphene. This makes the Dirac fluid in Sc-Herbertsmithite an interesting candidate for experimental realizations of holographic hydrodynamics.

**Robustness of the hydrodynamic regime.** We can understand the difference in the scales for Sc-Herbertsmithite as compared to graphene along the following lines. In order for an electron system to display hydrodynamic behavior, the electron-electron mean free path must obey[1]

$$\ell_{ee}(\alpha) \ll \ell_{imp}, \ell_{ph}, \ W, \tag{4}$$

where $\ell_{ph}$ is the electron-phonon mean free path, $\ell_{imp}$ is the electron-impurity mean free path, and $W$ the width of the sample. From Eq. (4), we infer that the emergence of hydrodynamic flow in solids is closely related to the relative value of the characteristic scales of the system[3,4]. This implies that the formation of viscous flow in solids is restricted to specific, and sometimes narrow, regimes. For Dirac fluids satisfying $\mu \ll k_BT$, the emergence of hydrodynamic behavior is greatly enhanced[2], since in this large temperature regime the available phase space for electron-electron collisions is vastly enhanced. In turn, Landau quasiparticles are short lived, which yields a decrease of $\ell_{ee}$. In addition, since $\ell_{ee} \propto 1/\alpha^2$, increasing the Coulomb interaction strength leads to shorter thermalization length and time scales. This leads to an enhancement of hydrodynamic behavior and a higher tendency to exhibit viscous flow[2]. Moreover, the emergence of additional non-hydrodynamic modes sets a critical length scale $\ell_c$, below which the standard hydrodynamic approach ceases to be valid. For the particular case of Sc-Herbertsmithite, we find (see "Methods" section)

$$\ell_c^{Hb} \simeq \frac{1}{6}\ell_c^{Gr}. \tag{5}$$

This shows that the hydrodynamic regime is more robust in Sc-Herbertsmithite than in graphene.

**Turbulent hydrodynamics**. Note that the shear viscosity controls the turbulent behavior of the fluid[22,23], an aspect of electron hydrodynamics that still remains largely unexplored. Within the hydrodynamic regime, the relative size of dissipative forces to nonlinear inertial forces in the fluid distinguishes the laminar flow regime, in which viscous forces dominate, from the turbulent flow regime, where non-linear dynamics in the Navier-Stokes equations dominate. More specifically, the ratio of inertial to dissipative/viscous forces is characterized by the Reynolds number of the flow. For a $2 + 1$ dimensional Dirac fluid at charge neutrality flowing within a channel of width $W$, the Reynolds number is given by

$$Re = \left(\frac{\eta}{s}\frac{k_B}{\hbar}\right)^{-1}\frac{k_B T}{\hbar v_F}\frac{u_{typ}(\eta/s)}{v_F}W, \qquad (6)$$

where $s$ the entropy density of the fluid and $u_{typ}$ is the typical velocity of the fluid, which strongly increases as $\eta/s$ decreases[24]. The fluid exhibits turbulent behavior when $Re = \mathcal{O}(1000)$[25].

The onset of turbulence is related to the fluid flow, which is sensitive not only to the sample geometry, but also to the intrinsic transport properties such as its shear viscosity—from Eq. (6), $Re \propto (\eta/s)^{-1} u_{typ}(\eta/s)$. Since the shear viscosity is much smaller for strongly coupled fluids (see Fig. 1), the experimental realization of turbulence will be greatly enhanced for materials exhibiting large Coulomb interactions. This conclusion holds regardless of the particular geometry employed. Furthermore, even laminar flows such as the Poiseuille channel flow will, compared to graphene or other even more weakly coupled materials, exhibit enhanced hydrodynamic transport properties such as larger typical fluid velocities and smaller differential resistances[24].

The third and final result of this work is an estimate of the Reynolds number for the flow of our proposed holographic Dirac fluid in Sc-Herbertsmithite through a long and straight channel flow as described by Eq. (6). In graphene, the Reynolds number has been found to be sufficiently large for preturbulent behavior such as vortex production, but not for fully developed turbulence[26]. For Sc-Herbertsmithite, we find an enhancement of the Reynolds number compared to graphene by a factor of order 100, mostly due to the Fermi velocity $v_F$ that is six times smaller (see Fig. 2b and Table 1, as well as "Methods" for the precise estimate of the enhancement factor), leading to Reynolds numbers sufficiently large for fully turbulent channel flows.

## Discussion

We hence predict that Sc-Herbertsmithite, and correlated Kagome metals at Dirac filling in general, bring the turbulent flow regime in two-dimensional electron hydrodynamics within experimental reach. As alternative routes, turbulent flow is predicted for three-dimensional hydrodynamic Weyl semimetals, from anisotropic two-dimensional Dirac fluids setting in at a possibly small Reynolds number[27,28], or particular superconductors in a fluctuation regime around $T_c$[29]. Interestingly, yet another promising pursuit of a turbulent electronic regime is to consider electronic fluids with particularly low entropy. This might be the case for the strange metal regime in cuprate superconductors[30].

We note that our theoretical results apply more generally than to the specific material candidate Sc-Herbertsmithite, since they provide a new viewpoint on viscous electron fluids in strongly correlated materials. Further examples they may apply to include copper-based metal-organic frameworks (MOFs) in which

electrons live on a Kagome lattice. These were recently synthesized successfully. They exhibit strong electronic correlations[31] and even superconductivity[32]. The fact that the number of available strongly correlated Kagome metals is growing significantly is encouraging in view of experimentally realizing our results, and outweighs the difficulties so far encountered in achieving conducting Herbertsmithite materials[33–35].

## Methods

**Properties of Herbertsmithite**. Kagome materials, such as Herbertsmithite combine the features of Dirac fermions and strong correlations. $ScCu_3(OH)_6Cl_2$, which we refer to as Sc-Herbertsmithite, consists of $CuO_4$ plaquettes forming a Kagome lattice (see Fig. 2a). The crystal field is such that the low-energy physics is correctly captured by a single $d_{x^2-y^2}$ orbital on each Cu site, and the resulting low-energy band structure is qualitatively similar to that of a one-orbital model at $n = 4/3$ electron filling, where, as shown in Fig. 2b, the Fermi level is pinned at the Dirac points.

As compared to graphene, where the underlying Dirac spectrum originates from weakly correlated $p_z$ orbitals, Sc-Herbertsmithite is expected to show a larger degree of electronic correlations. This is confirmed by our constrained random phase approximation (cRPA) estimates of the coupling strength $\alpha$. For a linear band dispersion, the strength of the Coulomb interaction is characterized by the effective fine-structure constant in Eq. (3). According to the values summarized in Table 1, $\alpha^{Sc-Hb} = 2.9 \pm 0.3$ as compared to $\alpha^{Gr} = 0.5 - 1.0$ for (hBN-encapsulated) graphene. We use $\alpha^{Gr} = 0.9$ as a representative example in the main text. The latter value was evaluated using an ab-initio estimate of the dielectric constant[36,37].

**First principles calculations**. For our numerical study of Sc-Herbertsmithite, we employed state-of-the-art first-principle calculations based on the density functional theory as implemented in the Vienna ab initio simulation package (VASP)[38] following the projector-augmented-plane-wave (PAW) method[39,40]. We use the generalized gradient approximation as parametrized by the PBE-GGA functional for the exchange-correlation potential[41] by expanding the Kohn-Sham wave functions into plane waves up to an energy cutoff of 400 eV and sampling the Brillouin zone on an $6 \times 6 \times 6$ regular mesh. We obtain the phonon dispersion $\omega_{\nu q}$ within the context of the density functional perturbation theory (DFPT)[42] as implemented in the Quantum ESPRESSO suite[43] with a $2 \times 2 \times 2$ supercell. The electron-phonon coupling strengths $\lambda_{\nu q} = \Pi''_{\nu q}/\pi N(\epsilon_F)\omega_{\nu q}$ are computed from the imaginary part $\Pi''_{\nu q}$ of the phonon self-energy within the Migdal approximation

$$\Pi_{\nu q} = \frac{1}{N_k}\sum_{mnk}|g^{k,q}_{mn,\nu}|^2 \frac{f(\epsilon_{nk}) - f(\epsilon_{mk+q})}{\epsilon_{mk+q} - \epsilon_{nk} - \omega_{\nu q} + i\eta} \qquad (7)$$

where $g^{k,q}_{mn,\nu}$ are the electron-phonon matrix elements and the electron momentum integration is performed on an extreme dense mesh of $N_k = 50^3$ points via Wannier interpolation[44].

We subsequently determine the Fermi velocity by a fit to the band structure along the $\Gamma$-X-J-$\Gamma$ path. The low-energy theory of Herbertsmithite consists of the Cu $d_{x^2-y^2}$ Dirac electrons and their hydrodynamic properties are characterized by the Coulomb coupling. This effective low-energy theory needs to take into account the effects of all other electronic states, in particular the dielectric screening. In other words, the hydrodynamic theory starts from partially dressed Cu $d_{x^2-y^2}$ electrons, with an effective dielectric constant $\epsilon_r$ that captures the screening by all other electronic states. Since these other states are far away from the Fermi level, they are less affected by correlation effects and the ab-initio calculation of their dielectric constant is feasible. To determine $\epsilon_r$, we start from the density functional theory band structure and fix (constrain) the occupation of all electronic states in the three Kagome bands to be $n = 4/3$ (including a factor of 2 from spin degeneracy), thereby excluding intraband screening processes[45]. To first approximation, the resulting dielectric tensor is diagonal and given by a single constant $\epsilon_r = 5$ (see Table 1), with relative deviations of up to 10% from this constant value. We use this 10% as an estimate of the methodological uncertainty. The cRPA dielectric constant and the Fermi velocity from density functional theory define the partially dressed electronic states entering the hydrodynamic theory, which captures the low-energy correlations in the material and determines the resulting viscosity.

**Phonons**. A number of caveats need to be addressed. Interactions of the electron fluid with lattice vibrations are detrimental to the hydrodynamic behavior. As shown in Fig. 2c, however, the optical phonon modes in Sc-Herbertsmithite that sensibly couple to the electronic states, and indeed contribute to the electron-phonon coupling, are populated for temperatures above $T_{ph} \sim 80$ K. This analysis indicates that the hydrodynamic regime can be expected to extend over a range of temperatures within current experimental reach.

**Instabilities**. The Kagome lattice itself plays a crucial role. In contrast to graphene's Dirac spectrum, which occurs precisely at half-filling, and as such it is prone to magnetic instabilities at comparably small couplings[20], the Dirac fluid in Kagome Sc-Herbertsmithite is reached at the stoichiometric filling $n = 4/3$. Therefore, tendencies towards a correlation driven magnetic groundstate are strongly suppressed, and the linear dispersion of the Dirac fluid is expected to be highly robust against the opening of a band gap. While the geometric frustration inherent in the Kagome lattice precludes a rigorous treatment, this conjecture is supported both from weak coupling[18,46] as well as strong coupling studies[19].

**Hydrodynamics**. Electronic fluids are in local thermal equilibrium on length scales larger than the interaction mean free path $\ell_{ee}$, which needs to be the smallest length scale for hydrodynamics to apply (see Eq. (4)). Locally, the laws of thermodynamics such as the Gibbs-Duhem relation

$$\epsilon + P = sT + \mu\rho \qquad (8)$$

holds for the densities of energy $\epsilon(T, \mu)$, entropy $s(T, \mu)$ and charge $\rho(T, \mu)$. Given the equation of state $P(\epsilon, \rho)$, Eq. (8) can be used to determine the entropy density $s$ ($\mu$, $T$). A two-dimensional Dirac fluid, i.e. a fluid of electrons with relativistic dispersion and the chemical potential pinned at the Dirac point, $\mu = 0$, is then described by relativistic hydrodynamics[22]. The hydrodynamic equations are the conservation equations of the energy-momentum tensor $T^{\mu\nu}$ and the electric current $J^\mu$,

$$\partial_\mu T^{\mu\nu} = F^{\nu\mu}J_\mu, \quad \partial_\mu J^\mu = 0, \qquad (9)$$

where we have also allowed for the coupling of the fluid to an external electromagnetic field, $F^{\nu\mu}$.

The hydrodynamic derivative expansion expresses $T^{\mu\nu}$ and $J^\mu$ order by order in derivatives of the local temperature $T(x^\nu)$, the chemical potential $\mu(x^\nu)$, and the relativistic 3-velocity $u^\mu(x^\nu)$ ($u^\mu u_\mu = -c^2$). For a parity and time reversal invariant, relativistic two-dimensional fluid, one obtains to first order in the derivatives

$$T^{\mu\nu} = T^{\mu\nu}_{(0)} + T^{\mu\nu}_{(1)}, \qquad (10)$$

$$T^{\mu\nu}_{(0)} = \varepsilon \, u^\mu u^\nu / c^2 + p\Delta^{\mu\nu}, \qquad (11)$$

$$T^{\mu\nu}_{(1)} = -\eta\Delta^{\mu\alpha}\Delta^{\nu\beta}\left(2\partial_{(\alpha u_\beta)} - \Delta^{\alpha\beta}\partial_\sigma u^\sigma\right) - \xi\Delta^{\mu\nu}\partial_\gamma u^\gamma, \qquad (12)$$

$$J^\mu = e\rho u^\mu + \sigma_Q\left(E^\mu - T\Delta_{\mu\nu}\partial^\nu \frac{(\mu/e)}{T}\right), \qquad (13)$$

where $\Delta^{\mu\nu} = u^\mu u^\nu/c^2 + \eta^{\mu\nu}$ is the projection matrix in the directions transverse to $u^\mu$ and $A_{(\mu\nu)} = (A_{\mu\nu} + A_{\nu\mu})/2$ denotes symmetrization of any two-tensor.

The shear viscosity $\eta$ is the main observable in this context. It can be calculated from kinetic theory in the perturbative regime $\alpha \ll 1$[47], and will be derived from a holographic model below in the nonperturbative regime $\alpha \gg 1$. The bulk viscosity $\xi$ is negligible due to the approximate scale invariance of the linear dispersion relation in a Dirac fluid, $\xi \approx 0$. For incompressible flows ($\partial_\mu u^\mu = 0$) such as the Poiseuille flow, a finite bulk viscosity has no effect either. The interaction-induced intrinsic conductivity $\sigma_Q$[48] affects electric transport and controls the rate of Joule heating, but does not affect the flow of the fluid.

The three transport coefficients ($\eta$, $\xi$, $\sigma_Q$) have to be calculated from a microscopic theory or an effective model, such as the kinetic theory at weak coupling or holography at strong coupling. In linear response theory around a global equilibrium (i.e., constant temperature $T$ and chemical potential $\mu$, and $u^\mu = (1, 0, 0)^T$), the shear viscosity $\eta$ can be calculated with the following Kubo formula at zero frequency $\omega$ and momentum **k**,

$$\eta = \lim_{\omega \to 0}\frac{1}{2i\omega}\langle[T_{xy}(\omega), T_{xy}(0)]\rangle. \qquad (14)$$

In the strong coupling regime, the AdS/CFT correspondence provides a framework to calculate $\eta$ via a semi-classical analysis of the dual gravitational action of the system[49,50]. The entropy density of the fluid is then given by the entropy density of the black hole horizon in the bulk of spacetime[51].

**Holographic model**. At infinite coupling and for a large number of degrees of freedom, the holographic dual is given by the Einstein-Hilbert action (in 2+1d dimensional field theory the coupling is dimensionfull and hence runs. This is usually reflected in holography by including a non-trivial dilaton scalar field, $\phi$, into the gravitational action. We have neglected the dilaton contribution in our analysis, because it leads to sub-leading derivative corrections to $\eta/s$)[52].

$$S_{EH} = \frac{1}{16\pi G_N}\int d^4 x \sqrt{-g}(R - 2\Lambda), \qquad (15)$$

yielding Eq. (1) for the ratio $\eta/s$, independently of the coupling constant and number of degrees of freedom. Our corrections to this limit are computed by adding higher powers of the curvature $R$ to Eq. (15). In $3 + 1d$ gravitational duals, the next-to-leading order $R^2$ terms contribute a topological Gauss-Bonnet term to the gravitational action[53] and therefore do not alter the value of $\eta/s$. Furthermore,

type-II supergravity, the ten-dimensional parent theory of our four-dimensional gravity dual, does not contain $R^3$ corrections, as was shown in ref. [54]. The next higher derivative corrections contain four powers of $R$[14,55,56] and are induced in type-II supergravity by quartic terms involving the Weyl tensor. These terms yield

$$\frac{\eta}{s} = \frac{\hbar}{4\pi k_B}\left(1 + \frac{\mathcal{C}'}{\lambda^{3/2}}\right). \qquad (16)$$

In top-down constructions of holography originating from string theory, the holographic dual theory is typically a non-Abelian gauge theory with gauge group rank $N$ and 't Hooft coupling $\lambda = \alpha N$, related to the fine-structure constant $\alpha$. $N$ parametrizes the number of degrees of freedom in the dual gauge theory. Dimensional analysis implies that the correction at order $R^4$ to $\eta/s$ scales as $\lambda^{-3/2}$ in any spacetime dimension, up to a multiplicative constant that depends on the details of the holographic model[49]. Eq. (16) in particular universally describes the coupling dependence of the ratio $\eta/s$ for all $(3 + 1)d$-holographic duals with relativistic symmetry at leading order in the inverse coupling expansion and in the large $N$ limit near charge neutrality. However, the prefactors of the allowed subleading curvature corrections are model dependent. We parametrize the model dependence of the $R^4$ correction through the prefactor $\mathcal{C}'$ in Eq. (16). For the original example of a theory with holographic dual, $\mathcal{N} = 4$ SYM theory[13], $\mathcal{C}' = 135\zeta(3)/8$[55]. The unknowns in Eq. (16) are $N$ and $\mathcal{C}'$, with $\mathcal{C}'$ depending on the particular holographic dual considered. We absorb $N^{-3/2}$ into $\mathcal{C}'$ via $\mathcal{C} \equiv \mathcal{C}'N^{-3/2}$, which brings Eq. (16) to the form of Eq. (2). Thus, the details of the holographic model suitable for describing Dirac fluids are encoded in a single coefficient, namely $\mathcal{C}$. Using the parametrization in terms of $\mathcal{C}$, we generate the blue band in Fig. 1 as follows: We vary $\mathcal{C}$ from $\mathcal{C} = 0.0005$ to $\mathcal{C} = 5$ to generate the blue band in Fig. 1. For $\mathcal{N} = 4$ SYM, this range of variation of $\mathcal{C}$ corresponds to formally varying $N$ from $N \simeq 10^3$ to $N \simeq 2$. As shown in Fig. 1, even varying $\mathcal{C}$ over four orders of magnitude changes the value of $\eta/s$ of Sc-Hb by at most a factor of two. In principle, there may be corrections of even higher order in $1/\lambda$ to Eq. (16), corresponding to terms involving even higher orders in the curvature. However, we do not expect these to alter the predicted estimate of the value of $\eta/s$ for values of the coupling not accessible to weak-coupling perturbation theory. Furthermore, the absence of $R^2$ and $R^3$ corrections to the gravitational action imply that Eq. (16) does not receive $1/N$ corrections[53]. Hence, corrections to $\eta/s$ independent of $\alpha$ can appear at order $1/N^2$ or higher, however, we expect these to be subleading as compared to the $\mathcal{C}/\alpha^{3/2}$ corrections at $\alpha^{Sc-Hb} = 2.9$, for the range of $\mathcal{C}$ chosen. Moreover, our main argument that the applicability of hydrodynamics and the appearance of turbulence are more likely at larger values of $\alpha$ is independent of the value of $N$. The black, fading, line in Fig. 1 shows the perturbative prediction at weak coupling[47,48], which cannot be extrapolated to stronger coupling without violating the condition $\eta \geq 0$ that follows from local application of the second law of thermodynamics

**Turbulence**. Using the extrapolation presented in Fig. 1, we infer that materials which display a stronger Coulomb interaction, such as Sc-Herbertsmithite, yield a much larger Reynolds number than that reported for graphene. In particular, using Eq. (6), Fig. 1 and the results for the Fermi velocity $v_F$ from Table 1, we find that Sc-Herbertsmithite will exhibit a 63 to 156 times larger Reynolds number as compared to graphene, depending on whether we use the value of $\eta/s$ at the bottom or the top of the blue band, respectively. This enhancement is large enough to expect fully developed turbulence in a constriction setup[26].

**Coupling dependence of the regime of validity of hydrodynamics**. In this section, we derive Eq. (5) for the relative ratio between the length scales where hydrodynamics is expected to break down for Sc-Herbertsmithite and graphene. One can characterize the breakdown of hydrodynamics by the existence of a diffusive pole, $\omega = -iDk^2$, in the energy-momentum tensor self-correlation in Eq. (14). Through the calculation of the poles of the mentioned self-correlation function via holography, the diffusive pole was shown[57] to disappear at a critical wavelength $\ell_c$ as the coupling strength of the system is decreased. The pole moves down the imaginary frequency axis and collides with an upward-moving pole at $\ell = \ell_c$. After the collision, the two poles split into a conjugate pair and the purely imaginary diffusion pole disappears. An approximate analytic result for $l_c$ is given by[57]

$$\ell_c \simeq \frac{hv_F}{0.04 k_B T}\lambda^{-3/2}, \qquad (17)$$

where $\lambda$ is the 't Hooft coupling defined below in Eq. (16). Within the model employed[57], the analytical approximation in Eq. (17) lies closer to the numerical value of $\ell_c$ for smaller values of $\lambda$. To our knowledge, $\ell_c$ was derived only through holographic methods. At weak coupling, it is only known that hydrodynamics has a finite radius of convergence. For the particular case of graphene and Sc-Herbertsmithite, with $\alpha^{Gr} = 0.9$ and $\alpha^{Sc-Hb} = 2.9$, we find the ratio given in Eq. (5). Note that we calculated only the ratio of critical wavelengths and not their absolute values. We did so because the two materials have the same low-lying energy spectra and thus Eq. (5) is independent of the unknowns $N, \mathcal{C}$ of Eq. (16). Eq. (5) shows that if both graphene and Sc-Herbertsmithite are described by holography, then the hydrodynamic approximation is more robust for Sc-Herbertsmithite. This

argument presumes that graphene can be described holographically. Because of graphene's intermediate coupling strength, however, $\alpha^{Gr} = 0.9$, it is more likely to lie in the intermediate regime, where neither simple truncated holographic models nor perturbative methods are applicable. In this case Eq. (5) entails that Sc-based Herbertsmithite is a better candidate for the realization of holographic hydrodynamics.

## Data availability
All data are available from the corresponding author upon request.

## Code availability
All codes used in this work, with the exception of the licensed VASP package, are either publicly available or available from the authors upon request.

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

### Acknowledgements

We thank Andy Lucas and Fabio Caruso for useful discussions. The work in Würzburg is funded by the Deutsche Forschungsgemeinschaft (DFG, German Research Foundation) through Project-ID 258499086—SFB 1170 and through the Würzburg-Dresden Cluster of Excellence on Complexity and Topology in Quantum Matter –ct.qmat Project-ID 390858490—EXC 2147. We acknowledge the Gauss Centre for Supercomputing e.V. for funding this project by providing computing time on the GCS Supercomputer Super-MUC at Leibniz Supercomputing Centre. Open access funding provided by Projekt DEAL.

### Author contributions

J.E., M.G., R.M., and R.T. initialized the project. J.E. and R.T. coordinated and supervised the investigation. I.M., R.M., and D.R.F. developed and performed the AdS-CFT analysis, while D.D.S., E.v.L., and T.W. performed the ab-initio and constrained RPA calculations. I.M., D.D.S., J.E., M.G., R.M., D.R.F., and R.T. wrote the paper. The paper reflects contributions from all authors.

### Competing interests

The authors declare no competing interests.
