## [Peer Review File · Nature Communications]

REVIEWERS' COMMENTS:

Reviewer #1 (Remarks to the Author):

I have read the revised version of the manuscript as well as the authors' responses to all three referee reports. These responses are detailed and I believe that the authors put a large amount of effort in addressing most of the issues raised to the best of their ability and to a very high standard. Naturally, limitations on what could be done (and be reasonably expected) exist because this paper crosses a number of subfields of physics (condensed matter physics, AdS/CFT, materials, ...). But this also makes the paper interesting and thought provoking. As a result of these detailed revisions, it is my opinion that the paper deserves to be published by Nature Communications.

Reviewer #2 (Remarks to the Author):

Although I feel the authors have done a competent job addressing most of the remarks of the previous referees, the central objection expressed in my previous report still remains. So, all I can do here is to expand on this point in greater detail.

Hydrodynamic behavior in the condensed matter physics context is unusual because of the special conditions needed for it to be realized. To date, evidence for this has only been found in materials exhibiting nearly ideal conduction electron behavior, such as graphene and PdCoO₂. In these materials, highly dispersive s-p electrons are involved in the conduction process. There are potentially other materials out there as well worthy of investigation. For instance, there are copper organics where dispersive conduction electrons are on a kagome lattice, and these materials even exhibit superconductivity (Huang et al, *Angew. Chem Int. Ed.* 57, 146 (2018)). So, had the authors picked one of these materials to study, I would have regarded this as a solid piece of work worthy of being considered for one of the Nature journals.

Instead, the authors chose to study the hypothetical material Sc-herbertsmithite. Certainly, at the time Mazin et al published their paper (Ref. 6), there was some optimism that perhaps conducting versions of herbertsmithite might exist. But recent work has dashed this hope. In the work of Kelly et al (*Phys. Rev X* 6, 041007 (2016)), almost two electrons per formula unit were introduced into herbertsmithite, yet the material remained insulating. In subsequent theory work by Liu et al (*Phys. Rev. Lett.* 121, 186402), this was understood to be due to polaron formation. And the one attempt to do as the authors suggested, which was to substitute the 2+ interlayer ions by 3+ ions by Puphal et al (*Phys. Status Solidi B* 256, 1800663 (2019)), also did not lead to conducting behavior. Although it is true that substitution by Ga was not complete (only 80%), the fact is, the best single crystals of herbertsmithite itself also have incomplete substitution (85%). Despite the authors' assertion about ionic sizes, the situation in the magnesium version of herbertsmithite is not that much different, despite the ionic size difference between Mg and Zn.

In retrospect, this should not have been a surprise. Unlike cuprates that exhibit mobile carriers (though it would be extremely doubtful that these would ever be in the hydrodynamic regime as well), most other 3d transition metal oxides do not. Cuprates are special in that they are strongly in the charge transfer regime, thus promoting mobile carrier formation. But this is not the case for the herbertsmithite class of materials. Because of the replacement of 2- oxygen by 1- ions (OH, Cl), the energetics are completely different. In particular, one finds a large band gap of order 4 eV in herbertsmithite, more comparable to NiO, say, than undoped cuprates. I would bet the authors an expensive bottle of wine that even if Sc-herbertsmithite could be synthesized, it will not be conducting. Because of this, I just don't see the rationale behind the present paper, and unfortunately cannot recommend its publication in Nature Communications. I say "unfortunately" because I personally found the paper to be very thought provoking.

Answer to Reviewers remarks

Reviewer #1 (Remarks to the Author):

I have read the revised version of the manuscript as well as the authors' responses to all three referee reports. These responses are detailed and I believe that the authors put a large amount of effort in addressing most of the issues raised to the best of their ability and to a very high standard. Naturally, limitations on what could be done (and be reasonably expected) exist because this paper crosses a number of subfields of physics (condensed matter physics, AdS/CFT, materials, ...). But this also makes the paper interesting and thought provoking. As a result of these detailed revisions, it is my opinion that the paper deserves to be published by Nature Communications.

We thank the referee for the positive assessment of our manuscript, as he/she recommends publication in Nature Communications.

Reviewer #2 (Remarks to the Author):

Although I feel the authors have done a competent job addressing most of the remarks of the previous referees, the central objection expressed in my previous report still remains. So, all I can do here is to expand on this point in greater detail.

We thank the referee for acknowledging our competent and detailed effort to address all criticism which had been raised in the course of the peer review. Regarding his/her central objection we will comment on below.

Hydrodynamic behavior in the condensed matter physics context is unusual because of the special conditions needed for it to be realized. To date, evidence for this has only been found in materials exhibiting nearly ideal conduction electron behavior, such as graphene and PdCoO₂. In these materials, highly dispersive s-p electrons are involved in the conduction process. There are potentially other materials out there as well worthy of investigation. For instance, there are copper organics where dispersive conduction electrons are on a kagome lattice, and these materials even exhibit superconductivity (Huang et al, Angew. Chem Int. Ed. 57, 146 (2018)). So, had the authors picked one of these materials to study, I would have regarded this as a solid piece of work worthy of being considered for one of the Nature journals.

First of all, we do not question at all that the experimental evidence for weakly correlated electron hydrodynamic behaviour is much richer and less ambiguous than for strongly correlated electron hydrodynamic behaviour. A central point of novelty contained in our work is precisely to enter this new territory. Not only is it a challenge to find a proper material, it is even harder to adopt a suitable theoretical modelling, which we accomplished through the AdS/CFT perspective on hydrodynamics. As such, there is substantial novelty contained in our manuscript beyond the specific material we propose, or rather use as a carrier for our theoretical ideas. We have accounted for the referees concerns by stating more explicitly that we expect our ideas to apply to an extended scope of materials classes, including copper organics, for which correlated electron hydrodynamics can be expected to be found.

Instead, the authors chose to study the hypothetical material Sc-herbertsmithite. Certainly, at the time Mazin et al published their paper (Ref. 6), there was some optimism that perhaps conducting versions of herbertsmithite might exist. But recent work has dashed this hope. In the work of Kelly et al (Phys. Rev X 6, 041007 (2016)), almost two electrons per formula unit were introduced into herbertsmithite, yet the material remained insulating. In subsequent theory work by Liu et al (Phys. Rev. Lett. 121, 186402), this was understood to be due to polaron formation. And the one attempt to do as the authors suggested, which was to substitute the 2+ interlayer ions by 3+ ions by Puphal et al (Phys. Status Solidi B 256, 1800663 (2019)), also did not lead to conducting behavior. Although it is true that substitution by Ga was not complete (only 80%), the fact is, the best single crystals of herbertsmithite itself also have incomplete substitution (85%). Despite the authors' assertion about ionic sizes, the situation in the magnesium version of herbertsmithite is not that much different, despite the ionic size difference between Mg and Zn.

In retrospect, this should not have been a surprise. Unlike cuprates that exhibit mobile carriers (though it would be extremely doubtful that these would ever be in the hydrodynamic regime as well), most other 3d transition metal oxides do not. Cuprates are special in that they are strongly in the charge transfer regime, thus promoting mobile carrier formation. But this is not the case for the herbertsmithite class of materials. Because of the replacement of 2-oxygen by 1-ions (OH, Cl), the energetics are completely different. In particular, one finds a large band gap of order 4 eV in herbertsmithite, more comparable to NiO, say, than undoped cuprates. I would bet the authors an expensive bottle of wine that even if Sc-herbertsmithite could be synthesized, it will not be conducting. Because of this, I just don't see the rationale behind the present paper, and unfortunately cannot recommend its publication in Nature Communications. I say "unfortunately" because I personally found the paper to be very thought provoking.

As commented on above, our theoretical ideas clearly transcend the specific material candidate at hand. Regarding the referee's criticism on Ga/Sc herbertsmithite, it is true that some unsuccessful attempts have been made to dope herbertsmithite, as the referee has correctly listed in the report. Still, the proof-of-principle synthesis of a Zn-free Ga/Sc only herbertsmithite type structure has not yet been attempted. While we thus agree that the existence of such a material remains to be investigated, our results apply far beyond this particular material. In the Discussion section of our revised manuscript, we now bring evidence of organic copper-based metal-organic frameworks as promising platforms to implement our theoretical proposal. The fact that the number of available strongly correlated Kagome metals is growing significantly is encouraging in view of experimentally realizing our results, and outweighs the difficulties so far encountered in achieving conducting herbertsmithite materials. We do hope that the adjustment of the main text, combined with this response also convinces the referee of our point of view.